# Matching Convolutional Neural Networks without Priors about Data

**Carlos Eduardo Rosar Kos Lassance**[1]**, Jean-Charles Vialatte**[1,2]**, Vincent Gripon**[1] **& Nicolas Farrugia**[1]
[1]IMT Atlantique, Brest, France: `firstname.lastname@imt-atlantique.fr`
[2]Cityzen Data, Brest, France: `firstname.lastname@cityzendata.com`

## Abstract

We propose an extension of Convolutional Neural Networks (CNNs) to graph-structured data, including strided convolutions and data augmentation on graph. Our method matches the accuracy of state-of-the-art CNNs when applied on images, without any prior about their 2D regular structure. On fMRI data, we obtain a significant gain in accuracy compared with existing graph-based alternatives.

## 1 Introduction

A main asset of CNNs ( LeCun et al. (1995)) is that they take advantage of the intrinsic regular 2D structure of the data. However when data lacks regular structure, there is no natural notion of convolutions, stride/pooling or data augmentation.

Methods to adapt CNNs to graphs have abundantly been proposed in the literature, e.g. Bruna et al. (2013); Duvenaud et al. (2015); Defferrard et al. (2016); Vialatte et al. (2016); Atwood & Towsley (2016); Kearnes et al. (2016); Monti et al. (2016); Kipf & Welling (2016); Niepert et al. (2016); Levie et al. (2017); Gilmer et al. (2017); Vialatte et al. (2017); Puy et al. (2017); Simonovsky & Komodakis (2017); Pasdeloup et al. (2017); Du et al. (2017); Velickovic et al. (2017); Hamilton et al. (2017); Tixier et al. (2017); Nikolentzos et al. (2017); Kondor et al. (2018); Li et al. (2018).

Contrary to many alternative works, we ensure that our proposed methodology matches the performance of CNNs when applied to regular domains even without any structure prior. The weight-sharing of our proposed convolutional layers are based on translations inferred upon a graph (given or inferred in case of no structure prior), as well as data-augmentation and stride.

Our method is able, without any structure prior, to reach similar performance to state-of-the-art CNNs on CIFAR10. We also demonstrate an accuracy gain on PINES fMRI, a neuroimaging dataset.

## 2 Methodology

Our method is based on Pasdeloup et al. (2017), where the authors have introduced a way to infer a graph from training signals, then translations from the obtained graph to design ad-hoc CNNs. Finding translations is an NP-complete problem such that for large graphs it is initially not practical. The authors proposed to find only local approximate translations and use them to move the kernel.

Here we extend this approach and design strided convolutions along graph downscaling, data augmentation and convolutions on downscaled graphs. As local translations can't be used to design the latter two, we design proxies to global translations. The outline of the method is depicted in figure 1.

### 2.1 Translations

We consider a graph $G = \langle V, E \rangle$ with $V$ the set of vertices, and $E \subseteq \binom{V}{2}$ the set of edges. $G$ is supposed to be connected, as conversely the process can be applied to each connected component.

**Definition 1** *A **candidate-translation** is a function $\phi : U \to V$, where $U \subset V$ and such that*
*1) $\phi$ is* injective*: $\forall v, v' \in U, \phi(v) = \phi(v') \Rightarrow v = v'$,*
*2) $\phi$ is* edge-constrained*: $\forall v \in U, (v, \phi(v)) \in E$,*
*3) $\phi$ is* strongly neighborhood-preserving*: $\forall v, v' \in U, (v, v') \in E \Leftrightarrow (\phi(v), \phi(v')) \in E$.*

Figure 1: Outline of the proposed method

The loss of $\phi$ is defined as $|V - U|$. Two candidate-translations $\phi$ and $\phi'$ are said to be aligned if $\exists v \in V, \phi(v) = \phi'(v)$. We define $N_r(v)$ as the set of vertices that are at most $r$-hop away from a vertex $v \in V$.

**Definition 2** *A **translation** in $G$ is a candidate-translation such that there is no aligned translation with a strictly smaller loss, or is the identity function.*

**Definition 3** *A **local translation** of center $v \in V$ is a translation in the subgraph of $G$ induced by $N_2(v)$, that has $v$ in its definition domain.*

**Definition 4** *A family of **proxy-translations** $(\psi_p)_{p=0,...\kappa-1}$ initialized by $v_0 \in V$ is defined algorithmically as follow: (we run several experiments while changing $v_0$ and keep the best result)*
*1) We place an indexing kernel on $N_1(v_0)$ i.e. $N_1(v_0) = \{v_0, v_1, ..., v_{\kappa-1}\}$ with $\forall p, \psi_p(v_0) = v_p$,*
*2) We move this kernel using each local translation $\phi$ of center $v_0$: $\forall p, \psi_p(\phi(v_0)) = \phi(v_p)$,*
*3) We repeat 2) from each new center until saturation. If a center is being reached again, we keep the indexing that minimizes the sum of losses of the local translations that has lead to it.*

For the sake of simplicity we omitted the case of approximate translations, for details we refer the reader to Pasdeloup et al. (2017).

## 2.2 EXTENDED CONVOLUTION LAYERS

The operation of the *extended convolution layer* centered on the vertex $v \in V$ is defined as:

$$\mathbf{y}_v = h\left(\sum_{p=0}^{\kappa-1} \mathbf{w}_p \mathbf{x}_{\phi_p(v)} + b\right) \quad \text{with} \quad \left\{ \begin{array}{ll} \phi_p(v) = \psi_p(v) & \text{if } \psi_p \text{ is defined on } v \\ \phi_p(v) = \bot \notin V & else \end{array} \right.$$

where $\mathbf{x}_\bot = 0$, $h$ is the activation function, and $b$ is the bias term.

### 2.3 EXTENDED DATA AUGMENTATION

We use proxy-translations to move training vectors, artificially creating new ones. Note that this type of data-augmentation is poorer than for images since no flipping, scaling or rotations are used.

### 2.4 EXTENDED DOWNSCALING LAYERS

#### 2.4.1 EXTENDED STRIDED CONVOLUTIONS

Given an arbitrary initial vertex $v_0 \in V$, the set of kept vertices $V_{\downarrow r}$ is defined inductively as follows:
**1)** $V_{\downarrow r}^0 = \{v_0\}$,
**2)** $\forall t \in \mathbb{N}, V_{\downarrow r}^{t+1} = V_{\downarrow r}^t \cup \{v \in V, \forall v' \in V_{\downarrow}^t, v \notin N_{r-1}(v') \wedge \exists v' \in V_{\downarrow r}^t, v \in N_r(v')\}$.

The output neurons of the extended convolution layer with stride $r$ are $V_{\downarrow r}$.

#### 2.4.2 EXTENDED CONVOLUTIONS ON THE STRIDED GRAPH

Using the proxy-translations, we move a localized $r$-hop indexing kernel over $G$. At each location, we associate the vertices of $V_{\downarrow r}$ with weights of the kernel, thus obtaining what we define as induced $\downarrow r$-translations on the set $V_{\downarrow r}$. In other words, when the kernel is centered on $v_0$, if $v_1 \in V_{\downarrow r}$ is associated with weight index $p_0$, we obtain $\phi_{p_0}^{\downarrow r}(v_0) = v_1$. Subsequent convolutions at lower scales are defined using these induced $\downarrow r$-translations.

## 3 EXPERIMENTS

We tested our method on CIFAR-10, a tiny images dataset (Krizhevsky (2009), using a PreActRes-Net18 (He et al. (2016)) architecture, swapping convolutions and strided convolutions for extended ones. We tested different combinations of graph support and data augmentation. Table 1 summarizes our results. In particular, it is interesting to note that results obtained without any structure prior (91.07%) are only 2.7% away from the reference using classical CNNs on images (93.80%).

Table 1: CIFAR-10 result comparison table.

| Support | MLP Lin et al. (2015) | CNN | Grid Graph | | Covariance Graph |
|---|---|---|---|---|---|
| | | | Defferrard et al. (2016) | Proposed | Proposed |
| Full Data Augmentation | 78.62% | **93.80%** | 85.13% | 93.94% | 92.57% |
| Data Augmentation - Flip | —– | 92.73% | 84.41% | 92.94% | 91.29% |
| Graph Data Augmentation | —– | —– | —– | 92.81% | **91.07%** |

The PINES dataset consists of fMRI scans on 182 subjects, during an emotional picture rating task.(Chang et al. (2015)) To generate our dataset, we fetched individual first-level statistical maps (from `https://neurovault.org/collections/1964/`) for the minimal and maximal ratings. Full brain data was masked on the MNI template and resampled to a 16mm cubic grid, in order to reduce dimensionality of the dataset while keeping a regular geometrical structure. Using a shallow architecture, Table 2 shows that our method improves over CNNs, MLPs and other graph-based networks. The code is available at `http://www.github.com/brain-bzh/MCNN`.

Table 2: PINES fMRI dataset accuracy comparison table.

| Graph | None | | Neighborhood Graph | |
|---|---|---|---|---|
| Method | MLP | CNN (kernel 1x1) | Defferrard et al. (2016) | Proposed |
| Accuracy | 82.62% | 84.30% | 82.80% | **85.08%** |

## 4 CONCLUSION

We proposed a new methodology that extends classical convolutional neural networks to irregular domains represented by a graph. We performed experiments and showed that our method is able to match performance of classical convolutional neural networks on images without explicit knowledge about the underlying regular 2D structure. It also significantly outperforms existing alternatives.

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
