# OpenReview forum: "Matching Convolutional Neural Networks without Priors about Data"
_ICLR.cc/2018/Workshop — Reject_

### Official Review · AnonReviewer3 · 2018-03-07
**Interesting extensions to Pasdeloup et al.'s flavor of graph convolutional nets, could use some more comparison experiments**

**Rating:** 7
**Confidence:** 4

**Review:**

The paper proposes some extensions to the method of Pasdeloup et al. (2017) (which includes some of the same authors), which can be used to construct convolutional neural networks on arbitrary graphs. These extensions include strided graph convolutions and data augmentation.

I think the title is a bit misleading in that respect, because to me it implies that the entire method of designing these models is newly proposed in this work, rather than a couple of extensions. It might be a good idea to make this more clear in the title.

The method is evaluated on the CIFAR-10 dataset, and demonstrated to perform comparably to a conventional convolutional baseline. This is actually an impressive result, as the graph convolutional network has a much weaker prior - although I would argue there is still a prior, contrary to what is claimed in the title ("without priors about data"). The fact that there is any convolutional structure to be found in the data at all is still used as prior knowledge.

The method is also evaluated on an fMRI dataset, where it improves performance.

I think the extensions that are introduced are valuable. While I realise that this is a workshop submission, I would still have appreciated a more thorough comparison to the state of the art - given that graph convolutional networks are intensively studied right now and there are many proposed implementations, as paragraph 2 of the introduction adequately shows. The claim in the conclusion that the method "[..] significantly outperforms existing alternatives" is not very well founded.

Figure 1 is great, as it allows the reader to get an idea of what the paper is about at a glance.

The notation in section 2.2 is a bit unclear, the symbol ⊥ seems to be used incorrectly at first glance.

The paper is well-written, and considering the length constraint, relatively easy to follow. The proposed extensions to the method of Pasdeloup et al. are interesting and the results are promising, so I think this would make a good workshop contribution.

---

### Official Review · AnonReviewer1 · 2018-03-09
**An extension of CNN to graph-structured data with strided convolution and data augmentation on graphs.**

**Rating:** 4
**Confidence:** 2

**Review:**

The paper presents an extension of the method proposed on Pasdeloup et al (2017) with strided convolutions along graph downscaling, data augmentation and convolutions on downscaled graphs. The method is poorly presented providing no intuition of the mathematical definitions, Figure 1that outlines the proposed method it not explained or any connection with the mathematical definitions and equations is provided. As far as the experiments section, no information is provided about the hyper-parameter tuning and the training-testing schema used in the experiments. Furthermore, it would be interesting to present also comparison with the Pasdeloup et al (2017) method.

---

### Official Review · AnonReviewer2 · 2018-03-09
**Key idea is not explained enough. Concerns about experiments.**

**Rating:** 4
**Confidence:** 4

**Review:**

This work seems to extend the prior work of Pasdeloup et al. (2017) to perform strided convolution. This strided convolution is only discussed in a very short Section 2.4 which doesn't provide much insight. From procedure in Sec. 2.4.1 it appears that authors sequentially add vertices of increasing (graph) distance from a center node. Such procedure does not down-sample as every subsequent stride will usually contain more vertices. If I understood correctly, that striding procedure is the main novelty, I think it should be given more attention in the paper.

Experiments:
In Table 1 accuracy for Full Data Augmentation with proposed procedure using Grid Graph (93.94) is higher than one achieved using CNN (93.8). This seems quite surprising and is not mentioned in the manuscript.

When comparing to Defferrard et al. (2016), I would be interested to see results on the data used in their paper (20news groups).

As the method is based on Pasdeloup et al. (2017), comparison to it seems necessary and is missing.

Henaff et al. [1] is one of the first practical methods generalizing CNNs to graphs. Considering it in the experimental section (and citing in the paper) could make the results more convincing.

[1] Mikael Henaff, Joan Bruna, and Yann LeCun, “Deep convolutional networks on graph-structured data,” arXiv preprint:1506.05163, 2015.

---

### Decision · Program_Chairs · 2018-03-20
**ICLR 2018 Workshop Acceptance Decision**

**Decision:**

Reject

**Comment:**

Based on the reviews, this paper has not been accepted for presentation at the ICLR workshop. However, the conversation and updates can continue to appear here on OpenReview.